# Wide Field of View Microwave Interferometric Radiometer Imaging

**Ignasi Corbella** [1,*] **, Francesc Torres** [1] **, Nuria Duffo** [1] **, Israel Duran** [1] **,**
**Veronica Gonzalez-Gambau** [2] **and Manuel Martin-Neira** [3]

[1] Remote Sensing Laboratory, Universitat Politecnica de Catalunya, c/ Jordi Girona 1–3,
08034 Barcelona, Spain; xtorres@tsc.upc.edu (F.T.); duffo@tsc.upc.edu (N.D.); israel.duran@tsc.upc.edu (I.D.)

[2] Department of Physical Oceanography, Institute of Marine Sciences (ICM), CSIC and Barcelona Expert
Center, Passeig Maritim de la Barceloneta, 37–49, 08003 Barcelona, Spain; vgonzalez@icm.csic.es

[3] European Space Research and Technology Centre, European Space Agency,
2200 AG Noordwijk, The Netherlands; Manuel.Martin-Neira@esa.int

* Correspondence: corbella@tsc.upc.edu; Tel.: +34-934-017-228

**Abstract:** In microwave interferometric radiometers with a large field of view, as for example the Microwave Imaging Radiometer with Aperture Synthesis (MIRAS) onboard the Soil Moisture and Ocean Salinity (SMOS) satellite, one of the major causes of reconstruction error is the contribution to the visibility of the brightness temperature outside the fundamental period, defined on the basis of reciprocal grids. A mitigation method consisting of estimating this contribution through the application of a brightness temperature model outside the fundamental period is proposed. The main advantage is that it does not require any a posteriori addition of artificial scenes to the reconstructed image. Additionally, a method to avoid the sophisticated matrix regularization and inversion techniques usually applied in microwave interferometry is presented. Image reconstruction algorithms are implemented on a minimum grid size in order to maximize their numerical efficiency. An improved method to apply an apodization window to the reconstructed image for reducing Gibbs oscillations is also proposed. All procedures are generally described considering the single polarization case and successively implemented applying the MIRAS layout in both its single polarization and full polarimetric modes. Results show similar performance of the proposed algorithm with respect to the nominal one applied by SMOS. All algorithms are implemented in the MIRAS Testing Software and have been successfully used for scientific studies by other teams.

**Keywords:** interferometric radiometry; image reconstruction; error correction

## 1. Introduction

Interferometric radiometers are passive imaging instruments whose operation is based on the Van Cittert–Zernike theorem. Their main advantage with respect to other kinds of radiometers is that they do not need moving elements to produce images, as these are entirely formed through data processing of the raw measurements. This technique was firstly proposed for earth observation in Le Vine et al. [1] and Ruf et al. [2]. One of the most representative examples is the Microwave Imaging Radiometer with Aperture Synthesis (MIRAS) [3,4] embarked on board the SMOS (Soil Moisture and Ocean Salinity) satellite [5], launched by the European Space Agency in 2009 and still providing useful geophysical data to the scientific community.

As originally derived for optical signals, the Van Cittert–Zernike theorem states that the mutual intensity of a radiation is the two-dimensional Fourier Transform of the intensity distribution across the source. In microwave radiometry terms, the visibility function is the two-dimensional Fourier

transform of the brightness temperature image. An inverse Fourier transform should then allow recovering of the brightness temperature from the calibrated visibility measurements. Nevertheless, in wide field of view instruments, those imaging an extended source covering most of the space in front of the antenna, there are non-negligible effects such as antenna patterns differences, obliquity factor, decorrelation, crosstalk and others that alter substantially this basic relation [6,7].

The visibility function is measured by cross-correlating all pairs of analytic signals collected by individual antennas. Assuming these ones evenly distributed on a fixed structure (as in the MIRAS case, shown in Figure 1) the visibility function becomes sampled at discrete points $(u, v)$ on a space-limited regular grid [8]. Since the visibility equation is ultimately a Fourier transform, the recovered brightness temperature is affected by aliasing in case the antenna separation fails to meet the Nyquist rate, which is usually the case. In addition, the fact that the visibility is limited in space (due to the instrument's finite size) is equivalent to having a spatial filter that sets the spatial resolution of the recovered map and produces ripples in sharp transitions (Gibbs effect). Moreover, the combined effects of antenna pattern differences and spatial decorrelation make the visibility equation depart from a simple Fourier transform, inducing errors even in the alias-free field of view [9].

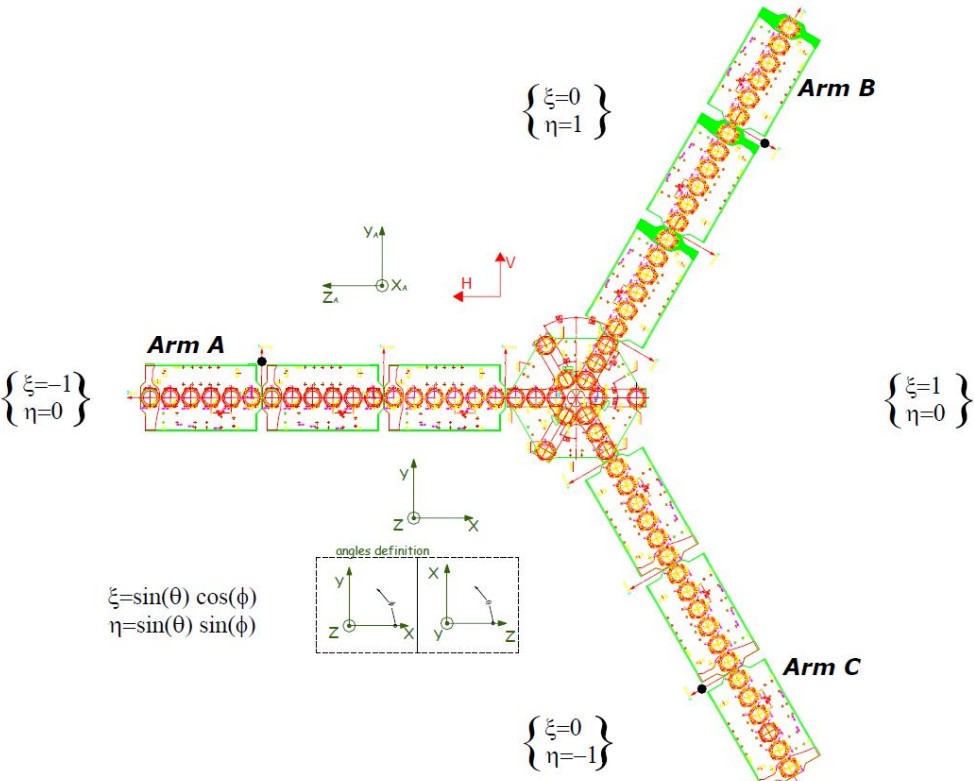

**Figure 1.** Microwave Imaging Radiometer with Aperture Synthesis (MIRAS) instrument layout and coordinate definition. Courtesy of AIRBUS Defence and Space [formerly EADS CASA Espacio].

The objective of this paper is to present an alternative image reconstruction algorithm for 2D interferometric radiometers. The algorithm is tested on a set of real measurements acquired by the MIRAS sensor, from which good results consistent with those obtained through the SMOS nominal processing chain are obtained. The paper is organized as follows: The main steps of the algorithm as well as MIRAS characteristics relevant to its application are described in Section 2; the results of the image reconstruction relying on the proposed algorithm are presented in Section 3; the discussion about these latter and those used by the SMOS nominal processing is given in Section 4; and finally the main conclusions are summarized in Section 5.

## 2. Methods

The main steps of the proposed image reconstruction algorithm are here presented by referring to the case of a single polarization measurement, and later extended to the case of a full polarimetric one. The algorithm is applied to the specific layout of the MIRAS radiometer (Figure 1). After the introduction of the visibility Equation (Section 2.1) and its conversion into a linear system of equations (Section 2.2), the algorithm develops through regularization of the matrix associated to the linear system of equations (Section 2.3), its inversion (Sections 2.4 and 2.5) and image reconstruction (Section 2.6). Lastly, the case of including the apodization into the processing chain is evaluated in Section 2.7 and the case of full polarimetric measurements is assessed in Section 2.8. Application to the MIRAS radiometer is illustrated throughout all the sections when needed.

### 2.1. Visibility Equation

The visibility equation to be used in aperture synthesis radiometry, derived in Corbella et al. [6], is a modified version of the Van Cittert–Zernike theorem to include the effect of the coupling between receivers and to fulfill the principle of energy conservation. In the single polarization case, after canceling the contribution of the receivers' physical temperature (i.e., approach 2 of Corbella et al. [8]) it is given by:

$$V(u,v) = \iint\limits_{\xi^2+\eta^2<1} T'(\xi,\eta)e^{-j2\pi(u\xi+v\eta)}d\xi d\eta, \tag{1}$$

where $T'(\xi,\eta)$ is the so-called "modified Brightness Temperature", expressed as:

$$T'(\xi,\eta) = T(\xi,\eta)\frac{F_k(\xi,\eta)F_j^*(\xi,\eta)}{\sqrt{1-\xi^2-\eta^2}\sqrt{\Omega_k\Omega_j}}\tilde{\bar{r}}_{kj}\left(-\frac{u\xi+v\eta}{f_0}\right), \tag{2}$$

in which $T(\xi,\eta)$ is the scene brightness temperature, $F_{k,j}$ are the complex field antenna patterns for the two elements $k$ and $j$, $\Omega_{k,j}$ is their corresponding antenna solid angles and $\tilde{\bar{r}}_{kj}(\ )$ is the normalized fringe washing function [6], which depends on the receivers' frequency responses. Only in the case of having identical antennas and neglecting the fringe washing function, the modified brightness temperature (Equation (2)) becomes independent of the specific antenna pair and Equation (1) reduces to a two-dimensional Fourier transform $V(u,v) = \mathscr{F}[T'(\xi,\eta)]$.

The domain variables for visibility $(u,v)$ and brightness temperature $(\xi,\eta)$ are defined as

$$\begin{aligned} u &= (x_j - x_k)/\lambda_0 & v &= (y_j - y_k)/\lambda_0 \\ \xi &= x/r & \eta &= y/r, \end{aligned} \tag{3}$$

where $(x,y)$ are the Cartesian coordinates of the observation point located at a distance $r$ from the instrument. This latter is assumed to be centered on the origin of coordinates and aligned with the $z = 0$ plane, with the antennas at coordinates $(x_{k,j}, y_{k,j})$ (see discussion below). Finally, $\lambda_0$ is the wavelength at the center frequency $f_0$, and $\xi$ and $\eta$ are the director cosines of the observation point with respect to axes $x$ and $y$ respectively, often expressed as a function of the elevation and azimuth angles $(\theta, \phi)$ as $\xi = \sin\theta\cos\phi$ and $\eta = \sin\theta\sin\phi$.

The antenna pattern of a given element $F_k(\xi,\eta)$ characterizes the electromagnetic field radiated by the whole structure when this particular element is active and no signal is applied to the rest. The antenna position $(x_k,y_k)$ in Equation (3) is the point at which its phase pattern is referenced to. To measure the embedded antenna pattern, the whole structure must rotate around a mechanical center of coordinates, so the radiated field becomes proportional to $F_{k0}(\xi,\eta)e^{-jkr}/r$ where $k = 2\pi/\lambda$ is the wave number and $F_{k0}(\xi,\eta)$ the pattern referenced to the coordinate center. The antenna phase pattern can then be referenced to any arbitrary position $(x_k, y_k, z_k)$ by expressing $r$ as $r = r_k + (r - r_k)$, with $r_k$ the distance from this position to the observation point. For large distances, the differential

length can be approximated by $r - r_k \approx \xi x_k + \eta y_k + \gamma z_k$ where $\gamma = z/r = \cos\theta$ is the third director cosine. The antenna pattern with phase referenced to coordinates $(x_k, y_k, z_k)$ is then

$$F_k(\xi, \eta) = F_{k0}(\xi, \eta)e^{-jk(x_k\xi + y_k\eta + z_k\gamma)}. \tag{4}$$

Using this equation, the "antenna position" $(x_k, y_k, z_k)$ can be chosen arbitrarily as long as the pattern phase is referenced to it. The position of the center of a sphere on which the phase variation of $F_k(\xi, \eta)$ is minimum is the antenna phase center, but this is not necessarily the best choice. In what follows, without loss of generality, the antenna positions are assumed to be equal to the nominal values and patterns are referenced to them. If antennas are properly designed, these positions should not be far away from their respective phase centers. And this is the case for SMOS.

In consequence the visibility (Equation (1)) is sampled at the the $(u, v)$ coordinates corresponding to the nominal antenna positions (Equation (3)) using the regular distribution of antennas within the instrument in the $z = 0$ plane, as shown in Figure 1 for the MIRAS case.

## 2.2. Discretization and G-Matrix

To solve Equation (1) the director cosine domain $(\xi, \eta)$ must also be discretized. The visibility equation becomes then a linear system of equations $V = GT$, where $G$ is a complex matrix whose elements are function of the individual antenna patterns [8], $V$ is the vector of visibilities in the $(u, v)$ space and $T$ the vector of brightness temperatures in the director cosine space $(\xi, \eta)$. The G-matrix, defined as linear operator relating visibility to brightness temperature, was originally proposed in Tanner et al. [10]. In principle, recovering the brightness temperature requires only inverting the system of equations: $T = G^{-1}V$. However, the G-matrix just defined happens to be ill-conditioned [11], so the solution is not straightforward.

The G-matrix has as many rows as visibility samples, including those corresponding to zero spacing (single antenna). For an instrument having N antennas there are $N(N-1)/2$ complex rows (MIRAS, with $N = 69$, has 2346) and as many real rows as number of antennas used to measure the antenna temperature (visibility at zero spacing). The current nominal SMOS processing uses only one, but there is a backup mode that uses all or a selected set of antennas for the visibility at the origin [12]. The number of columns of the G-matrix is the total number of grid points $(\xi, \eta)$ that fall inside the unit circle defined as $\xi^2 + \eta^2 < 1$.

Using Equations (1) and (2), the elements of the G-matrix are written as

$$G_{lm} = \Delta\xi\Delta\eta \, \frac{F_k(\xi, \eta)F_j^*(\xi, \eta)}{\sqrt{1 - \xi^2 - \eta^2}\sqrt{\Omega_k\Omega_j}} \, \tilde{\tilde{r}}_{kj}\left(-\frac{u\xi + v\eta}{f_0}\right) e^{-j2\pi(u\xi + v\eta)} \tag{5}$$

where the values of $(u, v)$ and $(\xi, \eta)$ are those of their respective grids and $\Delta\xi\Delta\eta$ is the elementary area (see Section 2.4).

## 2.3. Hermiticity and Redundant Baselines

Given the hermiticity property of the visibility $V(-u, -v) = V^*(u, v)$, for each complex row of the G-matrix an additional one can be added by changing the signs of $u$ and $v$, provided the corresponding row of the visibility vector is conjugated. This operation has to be performed before dealing with the redundant baselines.

Redundant baselines are those having identical $(u, v)$ values. Even though they correspond to the same visibility sample, they provide slightly different measurements with respect to each other because of the diverse antenna patterns of the involved elements. Considering two redundant baselines, the corresponding two distinct rows of the discretized visibility equation are:

$$V_l(u, v) = \sum_m G_{lm}T_m \quad ; \quad V_n(u, v) = \sum_m G_{nm}T_m, \tag{6}$$

where the subscripts $l$ and $n$ refer to two $(k, j)$ pairs of redundant baselines and the subscript $m$ ranges all $(\xi, \eta)$ pairs in the unit circle. Note that $(u, v)$ is the same in both by definition of redundant baselines. These two equations can be averaged to form a third one relating to the visibility of the same $(u, v)$ point to the scene brightness temperature

$$\bar{V}_l(u, v) = \sum_m \bar{G}_{lm} T_m, \tag{7}$$

where $\bar{V}_l(u, v) = (V_l(u, v) + V_n(u, v))/2$ and $\bar{G}_{lm} = (G_{lm} + G_{nm})/2$. This last equation can be used for inversion without any loss of information. As a matter of fact, different complete sets of visibility samples $V(u, v)$ are obtained by randomly choosing unique sets of non-redundant baselines. For each one, the corresponding visibility function becomes related to the same brightness temperature image, so the image reconstruction algorithm for each of them should yield the same result in the absence of noise and errors. So only one set of non-redundant visibilities is enough to fully recover the brightness temperature. Averaging all measurements of the same $(u, v)$ is not needed in the ideal case but has the effect of thermal noise reduction in practice.

The averaging operation must also be performed for the zero spacing visibility, which has a redundancy order equal to the number of antennas used to measure the antenna temperature.

After hermiticity extension and averaging of redundant visibilities, the number of rows of the G-matrix becomes equal to the total unique points in the $(u, v)$ domain. In MIRAS it is equal to 2791, of which one is real and the rest are complex. These two operations notably improve the G-matrix condition number, acting as a regularization method to make image reconstruction feasible. This method was used in both references [8] and [13], although these references do not mention it explicitly.

*2.4. Aliasing and Floor Error*

Since the visibility equation is fundamentally a Fourier transform, discretization grids for regular sampling in both domains must be reciprocal to each other. The lattice depends on the overall geometry: Rectangular for U-shaped instruments [14], or hexagonal [15] for Y-shaped (MIRAS), hexagonal [16] or triangular ones. In any case, the visibility sampling coordinates $(u, v)$ are a subset of the grid points in the fundamental period (a square for rectangular grids and a star or an hexagon for hexagonal grids). The minimum number of grid points in the fundamental period needed to include all measured samples is $N_T^2$ where $N_T = 4N_{EL} + 1$ for a hexagonal or triangular instrument, $N_T = 3N_{EL} + 1$ for a Y-shaped instrument or $N_T = 2N_{EL} + 1$ for a rectangular instrument [8]. In all cases, $N_{EL}$ is the number of elements in one arm. Since MIRAS has $N_{EL} = 21$, it follows that $N_T = 64$ in this case (The SMOS Level 1 Operational Processor uses $N_T = 196$). The number of points in the fundamental period of the corresponding $(\xi, \eta)$ reciprocal grid is also $N_T^2$. If reciprocal grids are used, the elementary area $\Delta\xi\Delta\eta$ in Equation (5) becomes then equal to $1/(N_T^2 d^2)$ for rectangular grids or $1/(N_T^2 d^2 \sin 60°)$ for hexagonal grids [8] where $d$ is the minimum antenna spacing normalized to the center wavelength.

Figure 2 shows the MIRAS reciprocal grids for $N_T = 64$. The fundamental hexagon is depicted in blue in both domains, the green star in $(u, v)$ includes all measured visibility points and their conjugate ones; and the extension of the $(\xi, \eta)$ grid to the unit circle is drawn in gray. Both fundamental hexagons have the same number of points, in this case equal to $64^2 = 4096$. A larger value just enlarges the $(u, v)$ hexagon while keeping the star shape identical, and provides a thicker grid in $(\xi, \eta)$ [8]. The fundamental hexagon in this domain has a fixed side, independent of the number of points, equal to $2/(3d)$ where $d$ is defined in the previous paragraph. This hexagon always falls inside the unit circle if $d > 1/\sqrt{3}$.

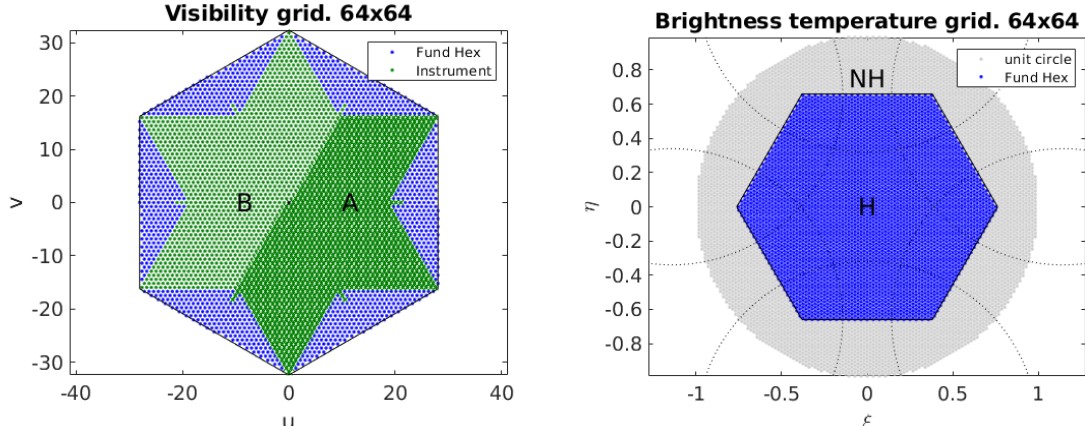

**Figure 2.** MIRAS reciprocal grids for visibility (**left**) and brightness temperature (**right**) using $N_T = 64$. Fundamental hexagons in both domains are drawn in blue. Two areas with hermitic $(u, v)$ points are highlighted in the left plot. Unit circles aliases are added in the right plot.

A Fourier transform needs zero padding to complete the $(u, v)$ fundamental period. The resultant modified brightness temperature is then obtained in the $(\xi, \eta)$ fundamental period. The periodic repetition of the phase produces the well known phenomenon of aliasing, resulting in that the same image is repeated at all adjacent periods. All unit circle aliases for the MIRAS case are depicted in Figure 2. The zone in which these circles do not overlap is the alias-free field of view. Out of it, the image is always contaminated with replicas of other areas of the same image. In principle, error-free imaging is only possible in the alias-free field of view unless the overlapped image content is null, as for example in the case of having a small target in the center of the field of view surrounded by a very low background. Contrarily, for wide field of view imagers, aliasing is a strong source of errors. Changing the normalized antenna spacing $d$ to a value lower than the Nyquist sampling rate ($1/2$ for rectangular grids and $1/\sqrt{3}$ for hexagonal grids) would make the circle be inscribed within the fundamental period, so eliminating the aliases.

For the minimum MIRAS reciprocal grid of Figure 2, the total number of $(\xi, \eta)$ points within the unit circle is $N_p = 8491$, so this is the number of columns of the G-matrix in this case. Splitting the columns into the fundamental hexagon $G_H$ and the rest of the unit circle $G_{NH}$ (see Figure 2), the discretized visibility equation can be written as

$$V = \begin{bmatrix} G_H & G_{NH} \end{bmatrix} \begin{bmatrix} T_H \\ T_{NH} \end{bmatrix} = G_H T_H + G_{NH} T_{NH}, \tag{8}$$

where the same nomenclature applies to $T_H$ and $T_{NH}$. Clearly, inverting only the G-matrix in the fundamental hexagon $T = G_H^{-1} V$ neglects the term $G_{NH} T_{NH}$ and produces what is sometimes called "floor error" [17,18]. Contrarily to the case of aliases in Fourier inversion, this one also spreads into the alias-free field of view unless considering identical antenna patterns and no fringe washing function. In this limiting case, imaging with G-matrix is equivalent to an inverse Fourier transform and the floor error is reduced to the aforementioned aliasing error.

In any case, the floor error can be mitigated by subtracting from the visibilities an estimation based on a forward a priori brightness temperature model outside the hexagon $M_{NH}$. Using Equation (8), the equation to invert becomes

$$V - G_{NH} M_{NH} \approx G_H T_H, \tag{9}$$

which leads to

$$T_H \approx G_H^{-1}(V - G_{NH} M_{NH}) = G_H^{-1} V - FE \, M_{NH}, \tag{10}$$

where $FE = G_H^{-1} G_{NH}$ is the floor error matrix. Even though computationally expensive, the floor error matrix is specific of the sensor and thus needs to be computed only once.

In conclusion: Image reconstruction is carried out by multiplying the inverse of the regularized G-matrix in the fundamental hexagon by the measured visibilities and substracting from the result an estimation of the floor error, which is equal to the product of the floor error matrix times a scene model outside the fundamental hexagon. Needless to say, the closer the model to the actual image, the lower the reconstruction error.

## 2.5. Matrix Extension and Inversion

Assuming that the regularization described in Section 2.3 has been applied and that the minimum reciprocal grids are used, the complex matrix $G_H$ has, in the MIRAS case, 2791 rows and 4096 columns, corresponding respectively to the $(u, v)$ unique grid points with measured visibilities (green star of Figure 2) and the fundamental hexagon in $(\xi, \eta)$. Its condition number is about 3.2. The matrix $G_H$ can thus be inverted using a Moore–Penrose pseudoinverse algorithm, as in Corbella et al. [8], by a conjugate-gradient method as in Camps et al. [13] or by other methods listed also in this reference.

A different approach is proposed here. First, the matrix is extended in the $(u, v)$ domain (rows) up to the whole principal hexagon (blue dots of Figure 2) using an average antenna pattern and unit fringe washing function. Using Equation (5), the G-matrix rows corresponding to the blue dots in Figure 2 (left), that is outside the star, are computed as

$$G_{lm} = \Delta\xi\Delta\eta \ \frac{|\bar{F}_n(\xi, \eta)|^2}{\sqrt{1 - \xi^2 - \eta^2}\Omega} e^{-j2\pi(u\xi + v\eta)}. \tag{11}$$

The extended G-matrix becomes then square with size $N_T^2 \times N_T^2$ and keeps the condition number, so it can be straightforwardly inverted with standard algorithms. Note that, if all antenna patterns were substituted by the average pattern and the fringe washing function was neglected, this extended G-matrix would be a Fourier matrix with all columns multiplied by the average antenna pattern. The product $G_H T$ would become then equivalent to the product of a Fourier matrix and the modified brightness temperature, as expected. The proposed method can be viewed as a modification of the Fourier inversion to include different antenna patterns. Although not specifically reported elsewhere, this inversion method has been successfully implemented in the MIRAS Testing Software [19] since its first version.

## 2.6. Image Reconstruction

The brightness temperature map is recovered by multiplying the calibrated visibility by the inverted extended $G_H$ matrix. For X or Y polarization the brightness temperature is real and $G_H^{-1}$ is hermitic, so the first term of Equation (10) can be written as

$$G_H^{-1}V = \left[ G_H^{-1}\big|_A \ \ G_H^{-1}\big|_0 \ \ G_H^{-1}\big|_B \right] \begin{bmatrix} V_A \\ V_0 \\ V_B \end{bmatrix} = G_H^{-1}\big|_0 V_0 + 2\Re e \left[ G_H^{-1}\big|_A V_A \right], \tag{12}$$

where the subscript 0 refers to the origin ($u = v = 0$) and $A$ and $B$ are two grid point subsets (matrix columns) having opposite $(u, v)$ signs. The ones used in the MIRAS Testing Software are shown in Figure 2 but the splitting is arbitrary. For points outside the star the visibility is ignored (see comment below about zero padding), so the corresponding columns of $G_H^{-1}$ are not used. The last equality in the above equation holds because of the hermiticity property of visibility function. Since both $G_H^{-1}\big|_0$ and $V_0$ are real, this equation can be written in a more compact form as

$$G_H^{-1}V = \Re e \left\{ \left[ G_H^{-1}\big|_0 \ \ 2G_H^{-1}\big|_A \right] \begin{bmatrix} V_0 \\ V_A \end{bmatrix} \right\}. \tag{13}$$

In MIRAS, using the minimum reciprocal grids, this operation involves the multiplication of a $4096 \times 1395$ complex matrix by a complex vector of 1395 elements.

If the complex polarimetric brightness temperature $T_{xy}$ is being imaged, the full $G_H^{-1}$ matrix should be used instead of its real part, although points outside the star should also be discarded.

The second term of Equation (10) does not depend on the measurement since it is computed using a model outside the hexagon. The hermiticity property of the $X$ and $Y$ polarizations can also be used to reduce the size of the floor error matrix in this case.

$$ FE = \Re e \left\{ \left[ \left. G_H^{-1} \right|_0 \;\; 2G_H^{-1} \right|_A \right] \left[ \begin{array}{c} \left. G_{NH} \right|_0 \\ \left. G_{NH} \right|_A \end{array} \right] \right\}. \tag{14} $$

The size of the MIRAS single polarization floor error matrix using the minimum size grids is always $4096 \times 4395$ corresponding to the $(\xi, \eta)$ points in both the fundamental hexagon and outside it respectively. This matrix is real for $T_x$ and $T_y$ and complex for $T_{xy}$.

A final comment about zero-padding is worth mentioning. The above equations detail the computation of each one of the two terms of Equation (10) separately but in a consistent way. Considering the version of this equation written at the first equal sign (that is $T_H = G_H^{-1}(V - G_{NH}M_{NH})$), it comes out that zero padding outside the star means using in these points the product $G_{NH}M_{NH}$, but not zero. In practice this needs not to be done explicitly, as it suffices just to ignore the columns of $G_H^{-1}$ outside the star.

### 2.7. Apodization

Due to the limited visibility coverage in the $(u, v)$ plane, the reconstructed brightness temperature is affected by the Gibbs effect showing ripples around abrupt changes in the original scene, as for example coastlines. As it is well known from Fourier imaging, ripples can be reduced at the expense of degrading spatial resolution by using an apodization window in the original domain. In Corbella et al. [8] the apodization window was directly applied to the measured visibilities, which is correct if a Fourier inversion is used but it is at least questionable for the G-matrix technique. A more rigorous approach is to window the Fourier components of the reconstructed brightness temperature image. In this case, the apodized brightness temperature is related to the reconstructed brightness temperature $T$ by

$$ T_{apodized} = \mathscr{F}^{-1}\{W\mathscr{F}\{T\}\}, \tag{15} $$

where $W$ is the window function, which in SMOS is always of Blackman type.

The reconstructed brightness temperature is defined in the $(\xi, \eta)$ fundamental hexagon, so its Fourier components, to which the window function is applied, are defined in the $(u, v)$ fundamental hexagon. The DFT operation involved in Equation (15) assumes implicitly that the reconstructed brightness temperature ($T$) is a periodic function in $(\xi, \eta)$ replicating itself in hexagons adjacent to the fundamental one. Since in typical SMOS images the earth disk is at the bottom of the hexagon and the sky at the top, there are abrupt changes at the border of the fundamental period that may induce ripples. To mitigate them, constant temperature levels are subtracted from the sky and earth zones so as to have a zero mean image. Specifically, the constant temperature subtracted to the sky pixels is computed as the median of the recovered image in them, while the value subtracted to the Earth pixels is computed so as to cancel the Fourier component at the origin. The consequent reduction of the contrasts within the image cause the minimization of the associated ripples. The constant temperatures are added back after Fourier inversion.

This approach is strongly inspired on the incremental visibility image reconstruction method proposed in Camps et al. [13]. In that case, however, the method was applied to visibilities instead to frequency components, but the idea is the same.

### 2.8. Full Polarimetric Case

Considering a baseline formed by two dual-polarization antennas, the full polarimetric discretized visibility equation [20] can be written in terms of G-matrices as

$$V_{xx} = \mathbf{G_{xx}^{RR}T_x} + G_{xx}^{CC}T_y + G_{xx}^{RC}T_{xy} + G_{xx}^{CR}T_{yx} \tag{16}$$

$$V_{yy} = G_{yy}^{CC}T_x + \mathbf{G_{yy}^{RR}T_y} + G_{yy}^{CR}T_{xy} + G_{yy}^{RC}T_{yx} \tag{17}$$

$$V_{xy} = G_{xy}^{RC}T_x + G_{xy}^{CR}T_y + \mathbf{G_{xy}^{RR}T_{xy}} + G_{xy}^{CC}T_{yx} \tag{18}$$

$$V_{yx} = G_{yx}^{CR}T_x + G_{yx}^{RC}T_y + G_{yx}^{CC}T_{xy} + \mathbf{G_{yx}^{RR}T_{yx}}, \tag{19}$$

where, for example, $G_{xy}^{RC}$ denotes the G-matrix computed according to Equation (5) using the Reference (co-polar) pattern of the X polarization antenna and the Cross-polar pattern of the Y polarization antenna. The terms marked in boldface are the dominant ones, in case of antennas with negligible cross-polar patterns, as considered in Equations (1) and (2).

Averaging redundant baselines, adding hermitic points and extending the matrix to the full hexagon is here carried out for each of the sub-matrices using the procedures detailed in Sections 2.3 and 2.5. Once this is done, the combination of the four equations can be written as $V = GT$ where now $G$ is a matrix with dimension $4N_T^2 \times 4N_p$, where $N_T^2$ is the total number of points in the fundamental hexagon and $N_p$ the number of points in the unit circle (4096 and 8491 respectively for the MIRAS minimum grid). The floor error mitigation through the application of a model outside the fundamental period can be carried out in the polarimetric case in exactly the same manner as done for a single polarization. The part of the extended G-matrix inside the fundamental period has now a size of $4N_T^2 \times 4N_T^2$ and, when inverted, provides as result a square matrix:

$$G_H^{-1} = \begin{bmatrix} IG_{11} & IG_{12} & IG_{13} & IG_{14} \\ IG_{21} & IG_{22} & IG_{23} & IG_{24} \\ IG_{31} & IG_{32} & IG_{33} & IG_{34} \\ IG_{41} & IG_{42} & IG_{43} & IG_{44} \end{bmatrix}, \tag{20}$$

where $IG_{ij}$ are submatrices of size $N_T^2 \times N_T^2$.

Applying the hermiticity property to the matrices of the first two rows using the procedures of Section 2.6 the following equations are obtained:

$$T_x = \Re e \left\{ \left[ IG_{11}\big|_0 \ \ 2\,IG_{11}\big|_A \right] \begin{bmatrix} V_{x_0} \\ V_{x_A} \end{bmatrix} + \left[ IG_{12}\big|_0 \ \ 2\,IG_{12}\big|_A \right] \begin{bmatrix} V_{y_0} \\ V_{y_A} \end{bmatrix} + \right. \tag{21}$$
$$\left. + \left[ IG_{13}\big|_0 \ \ 2\,IG_{13}\big|_A \right] \begin{bmatrix} V_{xy_0} \\ V_{xy_A} \end{bmatrix} + \left[ IG_{14}\big|_0 \ \ 2\,IG_{14}\big|_A \right] \begin{bmatrix} V_{yx_0} \\ V_{yx_A} \end{bmatrix} \right\},$$

$$T_y = \Re e \left\{ \left[ IG_{21}\big|_0 \ \ 2\,IG_{21}\big|_A \right] \begin{bmatrix} V_{x_0} \\ V_{x_A} \end{bmatrix} + \left[ IG_{22}\big|_0 \ \ 2\,IG_{22}\big|_A \right] \begin{bmatrix} V_{y_0} \\ V_{y_A} \end{bmatrix} + \right. \tag{22}$$
$$\left. + \left[ IG_{23}\big|_0 \ \ 2\,IG_{23}\big|_A \right] \begin{bmatrix} V_{xy_0} \\ V_{xy_A} \end{bmatrix} + \left[ IG_{24}\big|_0 \ \ 2\,IG_{24}\big|_A \right] \begin{bmatrix} V_{yx_0} \\ V_{yx_A} \end{bmatrix} \right\},$$

$$T_{xy} = IG_{31}V_x + IG_{32}V_y + IG_{33}V_{xy} + IG_{34}V_{yx}, \tag{23}$$

in which the rows of the inverted G-matrix outside the star are always discarded. Note that, by definition, $T_{yx} = T_{xy}^*$, so there is no additional equation for this term. Implementation wise it is found that this identity holds to the machine precision, which is a consistency indicator.

The floor error matrix is computed analogously to the case of single polarization measurements (Section 2.6) but using the larger G-matrix defined here. The result is a matrix four times the size of that for single polarization.

## 3. Results

The image reconstruction methodology outlined in the previous sections is fully implemented in the MIRAS Testing Software [19]. This tool is being systematically used for scientific studies as a flexible alternative to the nominal SMOS Level 1 data products and with successful results, as reported for example in González-Gambau et al. [21].

Exploiting the processing capabilities of this software, the proposed algorithm was tested relying on a set of real MIRAS measurements. Specifically, a SMOS orbit was processed in full polarimetric mode (Section 2.8) using the all-LICEF calibration approach [12]. Figure 3 shows the retrieved geo-located images at X- and Y-polarizations corresponding to a single snapshot acquired on 24 January 2019 at 21:44 UTC, when the satellite was passing over the coast of Australia during an ascending orbit. The Blackman window is applied using the methods in Section 2.7.

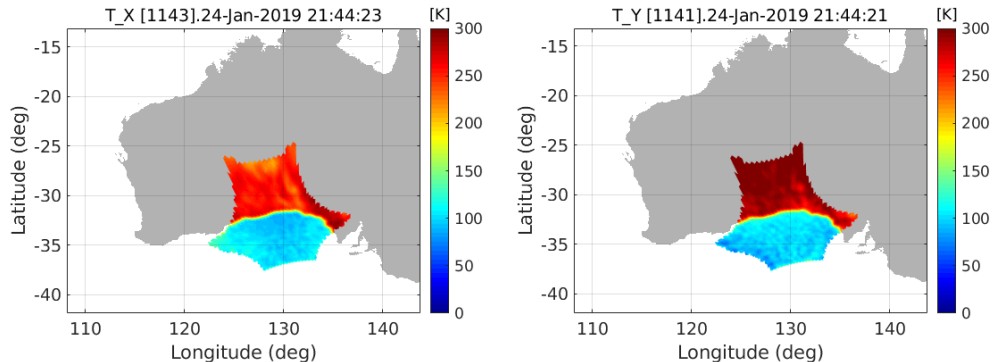

**Figure 3.** Geo-located images, in the extended alias-free field of view, of the two snapshots of Figure 4.

Figure 4 shows the brightness temperatures of same snapshot in the $(\xi, \eta)$ plane. Clearly, the image reconstruction algorithm is able to capture all the scene features in the whole hexagon, not only in the alias-free field of view. Ocean, land and sky areas are clearly distinguished and, at this scale, aliases impact is minor. Differences between $T_x$ and $T_y$ are due to the stronger $T_y$ increments associated to both the sea/land and Earth/sky transitions with respect to $T_x$. The model used to cancel the floor error [$M_{NH}$ in Equation (10)] consists of a constant value in land zones (258 K for X-pol and 285 K for Y-pol), specular reflection in ocean areas using Fresnel reflection coefficients with climatology salinity and temperature [22], a constant 8.5K to account for atmospheric effects and the L-Band sky and Galaxy map available as auxiliary file in the SMOS data base.

Figure 5 shows the X-polarization case with both terms of Equation (10) drawn separately as well as the difference between them, which is the final reconstructed image. The left panel shows the result of multiplying the calibrated visibility by the inverse of the G-matrix (first term of Equation (10), the center panel is the floor error and the right panel the corrected image. As expected, the floor error is very small in regions where the earth aliases do not enter into the hexagon, which is the so-called "extended alias-free field of view", nominally used in SMOS images (as in the geo-located images of Figure 3). Most of the artifacts of the raw image at the left are effectively removed in the corrected one. Specifically, strong improvement in the image reconstruction is found in the sky area as well as in both the bottom left and right strips. Contrarily, much lower effects are exhibited in the extended alias-free field of view, where the floor error is minimum. Later it will be shown that there is indeed a small improvement in this area. Apodization of the corrected image with a Blackman window, using the procedure of Section 2.7, provides the final result seen at the left of Figure 4.

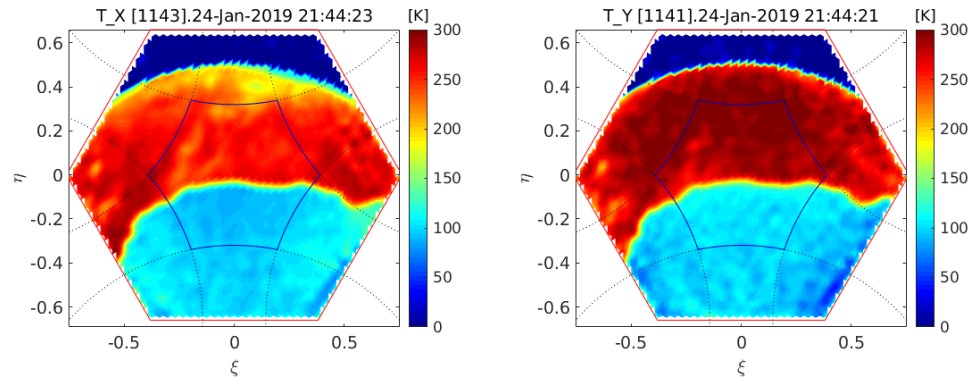

**Figure 4.** Image reconstruction result of two individual Soil Moisture and Ocean Salinity (SMOS) snapshots over the coast of Australia. Left is for X-pol and right is for Y-pol.

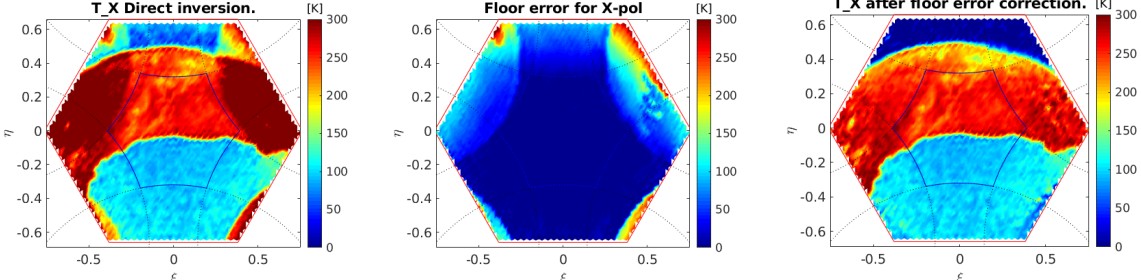

**Figure 5.** Floor error correction of the X-pol snapshot of Figure 4. Left and center images are first and second terms of Equation (10) respectively, and the difference is shown at right.

The scale of Figure 4 does not allow one to assess the effect of the floor error correction in the alias-free field of view. This is only possible using a differential image, plotting the brightness temperature bias with respect to its expected value. Additionally, averaging several snapshots is desirable in order to reduce thermal noise. Both requirements can be met if the scene is limited to snapshots over the ocean, for which a very comprehensive model is available from the SMOS science community (The authors would like to thank Joseph Tenerelli (OceanDataLab, France) for providing the ocean forward model).

Figure 6 shows the bias with respect to the model of all snapshots ranging in a latitudinal range of $[-40°, -5°]$ over the Pacific ocean in an ascending orbit of 28 January 2011 (chosen quite often for these kinds of analysis by the SMOS Level 1 team). In the area imaging the sky, the model is the standard SMOS Galaxy map. All four polarimetric products are included in Figure 6, brightness temperature at X and Y polarizations and real and imaginary parts of the complex brightness temperature. Basic statistics, referring to the alias-free field of view only, are shown at the bottom of each panel of Figure 6. Namely, spatial standard deviations and mean values are indicated by $\sigma$ and $T$ respectively. The relatively large negative bias in the Y-pol image may be due to poor modeling at high incident angles. In any case, these residual error images are compatible with the ones obtained with the SMOS Level-1 Operational Processor and have similar statistics.

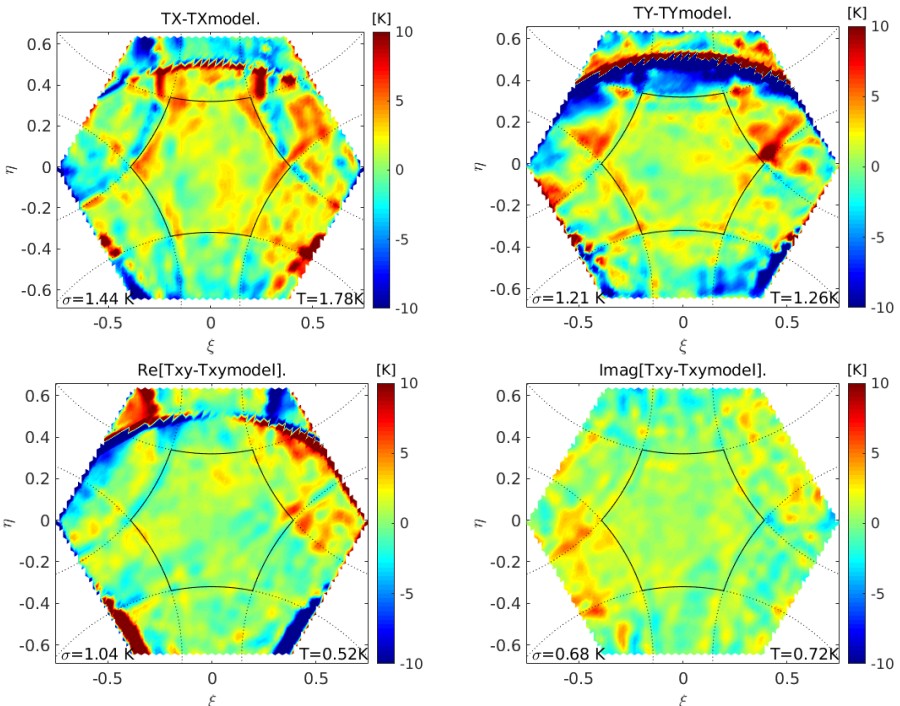

**Figure 6.** Spatial bias computed as the difference between the reconstructed brightness temperature and a model for the ocean. Blackman window is applied.

The same images have been produced without correcting the floor error, that is using only the first term of Equation (10). Results are shown in Figure 7. In this case, the error outside the alias-free field of view increases dramatically, especially in the areas in which the earth enters the hexagon. In the alias-free field of view there is a small impact in spatial bias, quantified in the standard deviation shown at the bottom. In all cases it increases with respect to the numbers provided in Figure 6. Note that the extended alias-free field of view is quite well recovered even if no correction is applied.

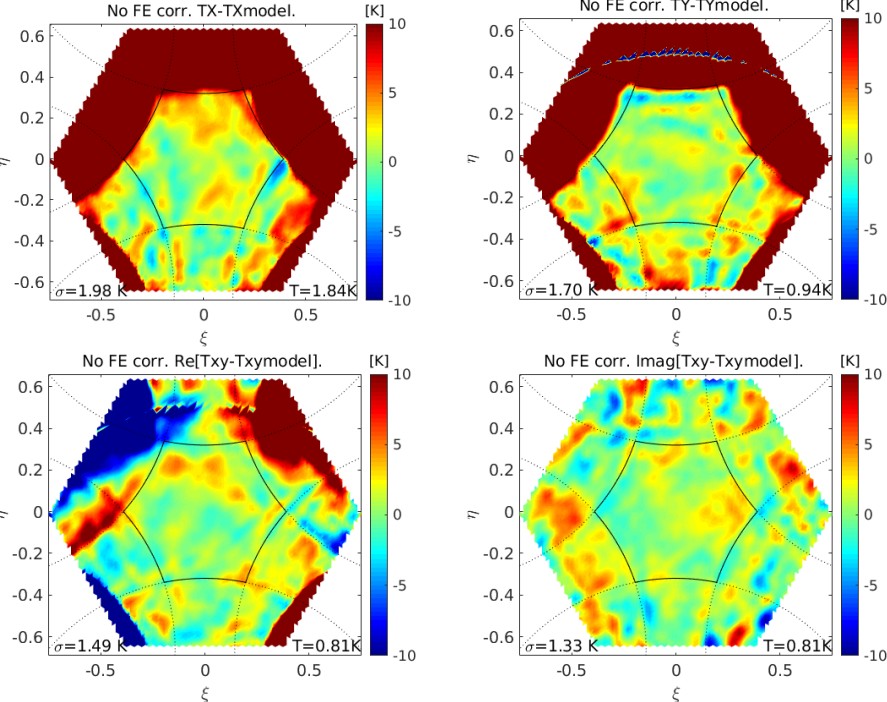

**Figure 7.** Spatial bias of Figure 6 without removing the floor error.

## 4. Discussion

　　Years before SMOS launch in November 2009, methods for 2D interferometric radiometer imaging were derived by different groups with the aim of having working algorithms as soon as visibility measurements were provided by MIRAS [8,11,23]. Radioastronomy heritage showed to be not directly applicable due to different instrument layouts, as for example the reduced antenna spacing, and especially because of the larger instantaneous field of view typical of earth observation. Working with simulated data it was early discovered that after a complete forward-backward simulation of a known scene, the original brightness temperature was not perfectly recovered even in the alias-free field of view. This misfit was called "scene-dependent bias" and was attributed to an underdetermination of the mathematical problem. An efficient mitigation algorithm was proposed in Corbella et al. [24], used in Anterrieu et al. [25] and improved later in Camps et al. [13]. This consists of subtracting from the visibility measurements different contributions estimated by simulation, and inverting the resultant "differential visibilities". Some of the contributions are later added back to the reconstructed image to recover the final brightness temperature map. The most recent implementation of this idea, described in Khazâal et al. [26], is included in the version 7 of the SMOS Level 1 Operational Processor. The method, even in its simplest version, is highly effective in canceling the sky aliases so expanding the alias-free field of view—limited by the unit circle aliases—to the extended alias-free field of view, limited by the earth shape. In its latest version [26], using a sophisticated model as "artificial scene" it is able to further reduce the error in the alias-free field of view.

　　All these methods are based on inverting only the portion of the G-matrix inside the fundamental hexagon discarding the rest. As already pointed out in Corbella et al. [17], discarding the G-matrix outside the fundamental hexagon is responsible for the appearance of the aliases. If all antenna patters were identical, this error would become limited to only the aliasing regions. In the real case, with different antenna patterns, the error is indeed larger in these regions but spreads also in the alias-free field of view (see center panel of Figure 5). The G-matrix in the whole unit circle is always used to estimate the corresponding visibility of the model or artificial scene, so imaging differential visibilities can be interpreted as removing an estimation of the aliases.

　　Using the concept of extended G-matrix of Section 2.5, it is easily shown that the method of imaging differential visibilities is equivalent to removing the floor error. The reconstructed brightness temperature from differential visibilities is

$$T = G_H^{-1}(V - GM) + M_H = G_H^{-1}V - (G_H^{-1}G - U)M, \tag{24}$$

where $U$ is a matrix with the same size of $G$ equal to the identity matrix for the columns inside the hexagon and zero outside $U = [I\ 0]$. In this equation, the G-matrix is extended, so having as many rows as number of points in the fundamental $(u, v)$ hexagon, and thus $G_H$ is a square and invertible matrix. The first term of this equation is the same as that of Equation (10). The second term can be expanded as

$$\left(G_H^{-1}\begin{bmatrix} G_H & G_{NH} \end{bmatrix} - \begin{bmatrix} I & 0 \end{bmatrix}\right)\begin{bmatrix} M_H \\ M_{NH} \end{bmatrix} = \begin{bmatrix} 0 & G_H^{-1}G_{NH} \end{bmatrix}\begin{bmatrix} M_H \\ M_{NH} \end{bmatrix} = G_H^{-1}G_{NH}M_{NH}, \tag{25}$$

which coincides with Equation (10). It is important to point out that this is only true if the visibility simulation of the model uses a G-matrix in the full unit circle defined in the same grid as the one used for inversion. That is, the matrix inverted is a subset of the complete one. Conceptually, a different matrix could be used since $V = GT$ is just a mathematical model of what the instrument actually measures. Also, the equivalence shown is based on the use of the extended G-matrix concept defined in Section 2.5. Different implementations of matrix inversion in Equation (24) are not equivalent. The main advantage of using the floor error matrix defined in Equation (10) is that processing does not

require estimating visibilities with the full G-matrix and does not require adding back any artificial scene to the result. It is a correction applied directly to the reconstructed image.

The outcome of the proposed methodology is the brightness temperature map of the scene. This is not the case for the SMOS Level 1 Operational Processor which, based in the proposal of Anterrieu et al. [11], defines the Level 1B data as the spatial frequency components of the brightness temperature. This is done through the concept of J-matrix, the concatenation of the G-matrix with the Fourier operator as dully explained in Khazâal et al. [26]. This approach is a different method to regularize the system of equations that relates all measured visibility, including redundant baselines, to brightness temperature. Due to redundancies the G-matrix in this case is ill-posed but the corresponding J-matrix is well behaved and can be inverted. As pointed out in Section 2.7, since the brightness temperature is not a periodic function, when recovering it from the corresponding frequencies, ripples can appear in the limits of the fundamental hexagon. In practice, this does not happen due to the imaging of differential visibilities used in the processor. The Level 1B data consists actually of frequency components of the difference between the brightness temperature and an artificial scene. After reconstructing the map including apodization, the latter is added back. Versions prior to 7 of the SMOS Level 1 Operational Processor use as artificial scene a constant in the earth disk and a sky map for the sky. In the end the constant earth is added to all points, and this is why the nominal SMOS images at Level 1B in the full hexagon show high temperature in the sky (see Figure 8). Here it has been shown that averaging redundant baselines improves the condition of the matrix and makes it invertible, so there is no need to compute the J-matrix. As a post-processing, however, the frequency components are computed in order to apply the Blackman window, as explained in Section 2.7.

For completeness, equivalent images as those shown in Figures 4 and 6 are provided respectively in Figures 8 and 9 using SMOS Level 1B data read from the ESA SMOS Online Dissemination Service, produced by version 6.21 of the SMOS Level 1 Operational Processor.

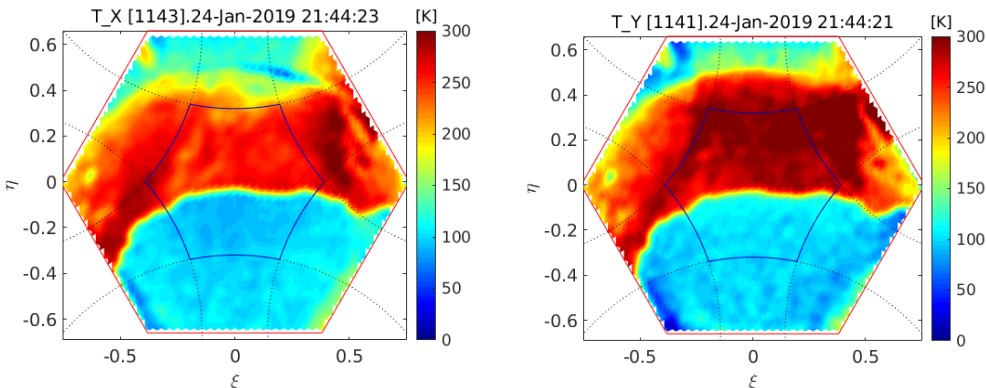

**Figure 8.** Same snapshots as in Figure 4 but using data from the SMOS Level 1 Operational Processor version 6.21. Deimos Engenharia, Lisbon (Portugal).

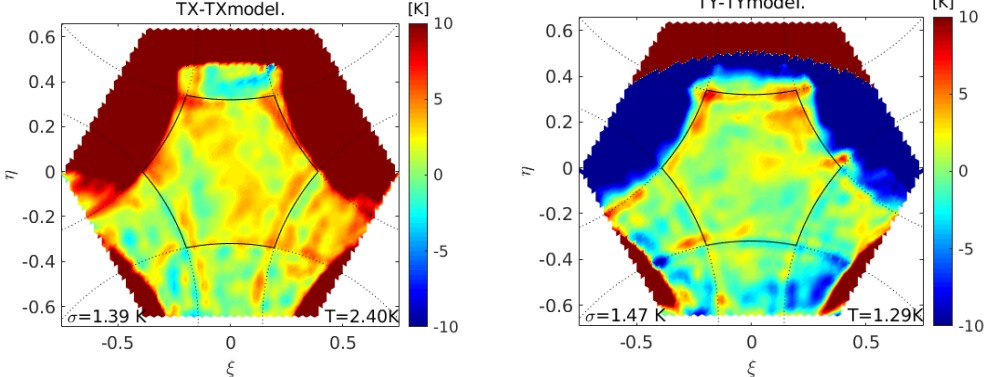

**Figure 9.** *Cont.*

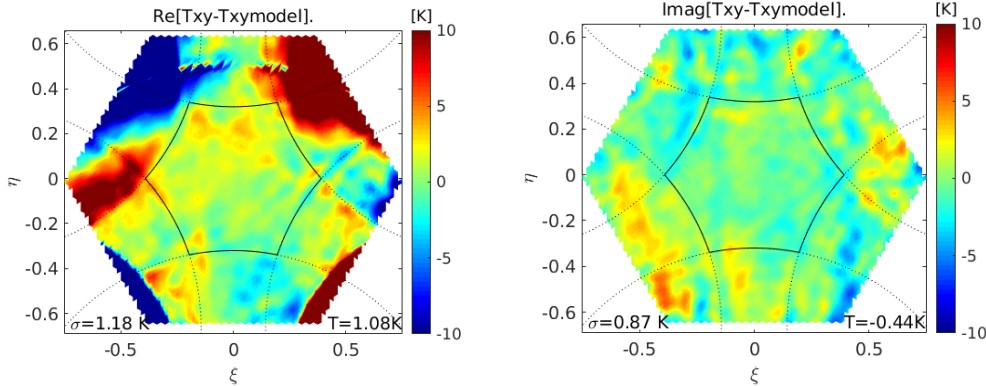

**Figure 9.** Same differential images as those in Figure 6 but using data from the SMOS Level 1 Operational Processor version 6.21. Deimos Engenharia, Lisbon (Portugal).

## 5. Conclusions

Accurate brightness temperature retrieval in 2D interferometric radiometers with a large field of view is not straightforward. The simplest approach consists of applying an inverse Fourier transform, which provides reasonable results in the alias-free field of view but large errors outside. The G-matrix approach takes into account individual differences between antenna patterns and also the fringe washing function and thus provides images of improved quality. In any case, the major error contribution is localized in the alias regions where brightness temperatures from several spatial directions overlap. For zones where the overlap is the sky the impact is low, but where aliases are produced by the earth it may become significant, affecting also the alias-free field of view. The origin of these errors is found in the contribution to the visibility of the brightness temperature from directions that fall outside the fundamental period. Subtracting an estimation of these visibilities to the measurements greatly reduces the error. Correction is carried out through the definition of the floor error matrix and relying on a brightness temperature model defined only outside the fundamental period.

Expressing the visibility equation as a linear system of equations, image reconstruction can be implemented through the inversion of the associated matrix, called G-matrix. Unfortunately, the G-matrix results are ill-conditioned and cannot be inverted without regularization. This is obtained by averaging redundant visibilities and extending the G-matrix to fill the whole hexagon in the $(u, v)$ plane, resulting in a square and well-conditioned matrix that is easy to invert. Using this approach, combined with the floor error removal, provides high quality images in the full hexagon, and especially in the extended alias-free field of view. The method is applicable to the full polarimetric operation of SMOS, with the only difference being increasing the size of the G-matrix. Results of SMOS complex images and full polarimetric error maps over ocean demonstrate the procedure.

The inversion method presented in this paper uses the minimum number of grid points, allowing for very fast and efficient programming. It has been implemented in the MIRAS Testing Software for several years and is being successfully used by science teams for salinity and soil moisture retrievals. In any case, the procedures outlined are not intended to replace the ones used by the SMOS Level 1 Operational Processor, but they are presented to the community with the objective of reporting other means of solving the same problem with similar results.

**Author Contributions:** Conceptualization, I.C.; Investigation, I.C. and I.D.; Methodology, F.T. and N.D.; Software, I.C.; Supervision, M.M.-N.; Validation, V.G.-G.; Writing—original draft, I.C.; Writing—review & editing, V.G.-G. and M.M.-N.

**Funding:** This research was funded by the European Space Agency through SMOS P7 subcontract DME CP12 no. 2015-005 with Deimos Enginheria (Portugal) and by Ministerio de Economia, Industria y Competitividad, Gobierno de España, projects TEC2014-58582-R, TEC2017-88850-R and ESP2015-67549-C3-1-R.

**Conflicts of Interest:** The authors declare no conflict of interest.

## Abbreviations

The following abbreviations are used in this manuscript:

SMOS    Soil Moisture and Ocean Salinity
MIRAS   Microwave Imaging Radiometer with Aperture Synthesis
DFT     Digital Fourier Transform

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
