# Peer review of "Wide Field of View Microwave Interferometric Radiometer Imaging"

_remotesensing, doi:10.3390/rs11060682_

Round 1

Reviewer 1 Report

The manuscript details the various steps of an alternative reconstruction image algorithm for interferometric radiometers with a wide field of view. The algorithm is tested on a set of real measurements acquired by the MIRAS sensor onboard SMOS, and compared to the method applied in the nominal SMOS processing chain.

The paper is generally well-written and effective in conveying its message. In addition, the fact of presenting in details such a widely used, but still unpublished algorithm makes it a relevant contribution to the literature on microwave radiometry.

Having said that, it can be improved in terms of readability and structure. Additionally, some of the figures can be reduced/simplified.

Namely,

- Introduction. Please add a description of the manuscript sections at the end of the introduction, as well as a description of the subsections at the beginning of section 2. Listing the various steps of the algorithm, or maybe including a little flow chart would clarify the processing chain and facilitate the reader.

- Methods. The manuscript would benefit from the creation of a specific subsection at the end of Methods dedicated to the description of MIRAS, or alternatively, to the application of the algorithm to the MIRAS case, including the instrument layout and all the N_* parameters at both single pol and full pol modes; figure 1 and 2 could be both moved to this section. This will permit a more general description of the algorithm, and at the same time will help the reader to fully understand the Results section.

- Results. The Authors might consider creating subsections in the Results, e.g., “Alias minimization”, “effects in the AF-FOV”.

- Figure 3. In my opinion, it can be eliminated, it does not add much information, and is actually mentioned in the paper only once to geolocate the snapshot.

- Figure 5. Considering the brief analysis of the figure, the Authors might consider to reduce it by showing only the Tx panel. Differently, some discussion about the different performance of the algorithm at the diverse polarizations must be included.

- Figures 6 and 7. As for figure 5, a more complete analysis of all the panels is needed.

- Discussion. The main objective of the discussion section is to present the comparison between alternative and nominal reconstruction algorithms. In this sense, the inclusion of a table summarizing common and different steps between the two reconstruction algorithms will certainly help the reader. Additionally, a figure showing the result of this comparison will be useful to support the discussion.

 Specific comments are included in the pdf file.

Author Response

RESPONSES ARE PROVIDED IN CAPITAL LETTERS

The manuscript details the various steps of an alternative reconstruction image algorithm for interferometric radiometers with a wide field of view. The algorithm is tested on a set of real measurements acquired by the MIRAS sensor onboard SMOS, and compared to the method applied in the nominal SMOS processing chain.

The paper is generally well-written and effective in conveying its message. In addition, the fact of presenting in details such a widely used, but still unpublished algorithm makes it a relevant contribution to the literature on microwave radiometry.

Having said that, it can be improved in terms of readability and structure. Additionally, some of the figures can be reduced/simplified.

Namely,

- Introduction. Please add a description of the manuscript sections at the end of the introduction, as well as a description of the subsections at the beginning of section 2. Listing the various steps of the algorithm, or maybe including a little flow chart would clarify the processing chain and facilitate the reader.

RESPONSE: THE TEXT SUGGESTED BY THE REVIEWER IN THE ANNOTATED pdf FILE HAS BEEN INCORPORATED, WITH SOME MINOR CHANGES TO AGREE ALSO WITH OTHER REVIEWERS.

- Methods. The manuscript would benefit from the creation of a specific subsection at the end of Methods dedicated to the description of MIRAS, or alternatively, to the application of the algorithm to the MIRAS case, including the instrument layout and all the N_* parameters at both single pol and full pol modes; figure 1 and 2 could be both moved to this section. This will permit a more general description of the algorithm, and at the same time will help the reader to fully understand the Results section.

RESPONSE: THE INTERLEAVING OF THE MIRAS CASE IN DEFINING THE ALGORITHMS IS ILLUSTRATIVE OF THEIR IMPLEMENTATION. MAKING A PURE FORMALLY MATHEMATICAL DESCRIPTION FIRST AND INCLUDING LATER MIRAS AS AN EXAMPLE CAN MAKE THE READER TO GO FORWARD AND BACKWARD TO UNDERSTAND THE DETAILS. FOR EXAMPLE, FIGURE 2 HELPS UNDERSTANDING EQUATIONS (8) AND (12), SO IT SHOULD BE KEPT IN ITS CONTEXT.
THE REVIEWER'S PROPOSAL REQUIRES A MAJOR CHANGE IN THE PAPER STRUCTURE, NOT ONLY MOVING AROUND TEXT BUT ALSO CHANGING THE WAY THINGS ARE EXPLAINED. OTHER REVIEWERS HAVE NOT QUESTIONED ABOUT THE STRUCTURE, SO IT IS THE DECISION OF THE AUTHORS NOT TO ENDORSE THE REVIEWER REQUIREMENT.    

- Results. The Authors might consider creating subsections in the Results, e.g., “Alias minimization”, “effects in the AF-FOV”.

RESPONSE: THE RESULTS ARE BASED ON ANALYZING GLOBALLY THE IMAGES GIVEN IN FIGURES 3 TO 6. NO SPECIFIC CORRECTIONS FOR ALIAS MINIMIZATION OR FOR AF-FOV ARE GIVEN, SO CUTTING THE TEXT INTO DIFFERENT SMALL SECTIONS WOULD NOT ADD INFORMATION AND WOULD BE MISLEADING.

- Figure 3. In my opinion, it can be eliminated, it does not add much information, and is actually mentioned in the paper only once to geolocate the snapshot.

RESPONSE: THIS FIGURE IS USEFUL TO UNDERSTAND THE IMAGES SEEN IN FIGURE 4. IN ANY CASE IT DOES NOT HARM, SO IT IS KEPT.

- Figure 5. Considering the brief analysis of the figure, the Authors might consider to reduce it by showing only the Tx panel. Differently, some discussion about the different performance of the algorithm at the diverse polarizations must be included.

RESPONSE: WHAT IS THE POINT OF SHOWING ONLY THE Tx PANEL?. THERE IS SPACE FOR BOTH AND THEY CAN BE VIEWED AS TWO DIFFERENT EXAMPLES OF IMAGE RECONSTRUCTION. THE ALGORITHM IS THE SAME IN BOTH POLARIZATIONS AND DOES NOT HAVE DIFFERENT PERFORMANCE.

- Figures 6 and 7. As for figure 5, a more complete analysis of all the panels is needed.

RESPONSE: DETAILED EXPLANATIONS OF THESE FIGURES ARE GIVEN IN THE TEXT THAT REFERENCES THEM. WHAT THE REVIEWER FOUNDS LACKING TO HAVE A "COMPLETE ANALYSIS"? PLEASE BE MORE EXPLICIT.

- Discussion. The main objective of the discussion section is to present the comparison between alternative and nominal reconstruction algorithms. In this sense, the inclusion of a table summarizing common and different steps between the two reconstruction algorithms will certainly help the reader. Additionally, a figure showing the result of this comparison will be useful to support the discussion.

RESPONSE: THE DISCUSSION SECTION IS NOT INTENDED TO MAKE A THOROUGH COMPARISON BETWEEN IMAGE RECONSTRUCTION ALGORITHMS. IT STARTS WITH A HISTORICAL PERSPECTIVE IN THE CASE OF SMOS AND PROVIDES A HINT ON WHAT IS IMPLEMENTED IN THE SMOS LEVEL 1 OPERATIONAL PROCESSOR. THEN IT SHOWS MATHEMATICALLY THAT THE METHOD PROPOSED IS EQUIVALENT TO THE ONE IMPLEMENTED IN THE LATEST VERSION OF THIS PROCESSOR. TO OUR UNDERSTANDING THERE IS NO REASON TO UNDERGO A DETAILED COMPARISON AMONG DIFFERENT METHODS. THIS MATHEMATICAL DEMONSTRATION SUFFICES. IN ANY CASE, FIGURES EQUIVALENT TO 4 AND 6 BUT USING DATA DOWNLOADED FROM SMOS ARE NOW INCLUDED FOR REFERENCE AND COMPARISON.

Specific comments are included in the pdf file.

RESPONSES TO ALL OF THEM ARE GIVEN ALSO IN THE pdf FILE

Reviewer 2 Report

The manuscript can be accepted in the present form. Well done!

Author Response

NO ITEMS TO RESPOND FROM THIS REVIEWER

Reviewer 3 Report

This paper presents a variation of the basic “G-matrix” approach used to create images from the “visibilities” measured by interferometric radiometers such as SMOS.  The authors claim that their approach (including averaging redundant baselines and removing a theoretical scene approximating the average scene) results in a more efficient, but equivalent, approach compared to the current SMOS official “processor”.

The manuscript is well written and readable.  There are a few cases where poor choice of words obscures meaning.   Some examples and typos:

a.  It is common in Europe to say “subtracted to” but “subtracted from” and “added to” sound better to this native, US English, speaker.

b.  Line 289:  “regions were” probably should be “regions where”

c.  Line 330:  “an a priori”

d.  Line 343:  “responsible for aliases apparition”  Meaning not clear.

e.  Line 387:  “and [also] the fringe”  Adding “also” clarifies the meaning.

f.  Line 390:  “earth disk” ?

g.  Line 400:  “and especially”?

h. Line 406:  “to substitute the ones” probably should be “to replace the ones”.

i.  Line 86 “inventing”

j. Line 343:  “aliases of apparition”  meaning not clear.

It is requested that the authors consider the following issues with the manuscript: 

1.  The examples presented to document the performance of the proposed approach (e.g. Figs 4-6) were not particularly convincing because there is no surface ”truth” to compare them to, and because the spatial and brightness temperature scale, even for the difference images, is coarse and hides artifacts. As a result, the statement in the conclusions (Line 402), “error maps over the ocean demonstrate the procedure” did not convince this reader.

2.  The phrase “Wide Field of View” in the title confused this reader.  It is the understanding of this reviewer that the problems of image reconstruction described here (image noise and aliasing) will be present from an Earth-viewing “interferometric” radiometer regardless of its field of view.  Perhaps an interferometer imaging a point source against a perfectly cold sky will not have these problems, but any such radiometer imaging the earth from space will have them whether or not it has a “wide field of view”. 

3.  “alias free” region:  It is the understanding of this reader that once aliasing is present and the scene fills the field of view of the antenna, there is no truly “alias free” region.  Even a very good antenna array observing at nadir will have error due to aliasing if the spacing is not less than 0.5 wavelengths.  How is SMOS’s “alias free” region different?

4.  “extended alias-free region”:  Lines 289-291 appear to be the only definition of this term, but they are not clearly written.  The phrase “which conform the” probably involves a typo.

5.  Averaging: 

a)  Lines 103-112:  The subject of the text is averaging rows in the matrix expression  V = GT with redundant spacing, but the wording is awkward.

b)  It is not clear to this reader that Eqn 7 can be used (line 112) “without any loss of information”.  Averaging in the manner suggested makes sense but it is a choice of options and it remains to be demonstrated in the case of different antenna patterns and variation in receiver characteristics, that there is no loss of information and that it results in thermal noise reduction (Line 118-119).

6.  Lines 139-140:  “hermitic ones”  Should this be “conjugate ones”?

7.  Lines 295-296:  Is figure 4 the correct figure?  There are only two examples in Fig 4.

8.  Eqn 4:  It is not clear to this reviewer that adding phase shift information to the antenna patterns is not counting the displacement of the receivers in a correlation pair twice.  Please check that in deriving Eqn 1, the assumption is made that the antenna pattern is at the receiver (i.e. in coordinates local to the individual receivers).

Author Response

RESPONSES ARE PROVIDED IN CAPITAL LETTERS.

This paper presents a variation of the basic “G-matrix” approach used to create images from the “visibilities” measured by interferometric radiometers such as SMOS.  The authors claim that their approach (including averaging redundant baselines and removing a theoretical scene approximating the average scene) results in a more efficient, but equivalent, approach compared to the current SMOS official “processor”.

The manuscript is well written and readable.  There are a few cases where poor choice of words obscures meaning.   Some examples and typos:

a.  It is common in Europe to say “subtracted to” but “subtracted from” and “added to” sound better to this native, US English, speaker.

ALL INSTANCES CHANGED

b.  Line 289:  “regions were” probably should be “regions where”

CHANGED

c.  Line 330:  “an a priori”

CHANGED

d.  Line 343:  “responsible for aliases apparition”  Meaning not clear.

CHANGED TO "responsible for the appearance of the aliases". MORE DETAILS ARE GIVEN IN THE CITED REFERENCE [15]

e.  Line 387:  “and [also] the fringe”  Adding “also” clarifies the meaning.

INCLUDED

f.  Line 390:  “earth disk” ?

CHANGED

g.  Line 400:  “and especially”?

CHANGED

h. Line 406:  “to substitute the ones” probably should be “to replace the ones”.

CHANGED

i.  Line 86 “inventing”

CHANGED TO inverting

j. Line 343:  “aliases of apparition”  meaning not clear.

SEE RESPONSE OF COMMENT d.

It is requested that the authors consider the following issues with the manuscript:

1.  The examples presented to document the performance of the proposed approach (e.g. Figs 4-6) were not particularly convincing because there is no surface ”truth” to compare them to, and because the spatial and brightness temperature scale, even for the difference images, is coarse and hides artifacts. As a result, the statement in the conclusions (Line 402), “error maps over the ocean demonstrate the procedure” did not convince this reader.

RESPONSE: ALL IMAGES DO USE A GROUND "TRUTH". FIGURE 4 USES A LAND-SEA MASK AND FIGURE 6 USES AN OCEAN EMISSION MODEL. FIGURE 5 IS INCLUDED ONLY TO ILLUSTRATE EQUATION (10), NOT TO PROVIDE RECONSTRUCTION ERRORS. THE STATEMENT IN THE CONCLUSION REFERS TO FIGURE 6 AS COMPARED TO FIGURE 7, WHICH BOTH COMPARE RESULTS WITH AND WITHOUT CORRECTION. SO THE PROPOSED IMAGE RECONSTRUCTION METHOD IS INDEED DEMONSTRATED BY THE IMAGES SHOWN, BOTH MIXED SCENES AND OVER THE OCEAN.
THE ARTIFACTS STILL SEEN IN FIGURE 6 COME FROM INSUFFICIENT KNOWLEDGE OF THE ANTENNA PATTERNS (AND PROBABLY OTHER UNKNOWN SOURCES) AND ARE FULLY CONSISTENT WITH THE ONES OBTAINED WITH THE NOMINAL SMOS PROCESSOR. AT THE END OF THE PAPER IMAGES USING DOWNLOADED SMOS DATA ARE NOW INCLUDED FOR COMPARISON.

2.  The phrase “Wide Field of View” in the title confused this reader.  It is the understanding of this reviewer that the problems of image reconstruction described here (image noise and aliasing) will be present from an Earth-viewing “interferometric” radiometer regardless of its field of view.  Perhaps an interferometer imaging a point source against a perfectly cold sky will not have these problems, but any such radiometer imaging the earth from space will have them whether or not it has a “wide field of view”.

RESPONSE: THE TERM “Wide Field of View” REFERS TO HAVING AN EXTENDED SOURCE COVERING VIRTUALLY ALL THE SPACE IN FRONT OF THE ANTENNA. THIS HAPPENS ALWAYS WHEN IMAGING THE EARTH FROM A LOW ORBIT SATELLITE (OR AN AIRPLANE). OF COURSE, FOR A SPECIFIC APPLICATION THE USABLE FIELD OF VIEW MAY BE REDUCED TO ONLY A SMALL PORTION OF THE IMAGE, BUT THIS PARTICULARITY DOES NOT CHANGE THE METHOD. A CLARIFICATION HAS BEEN INCLUDED IN THE INTRODUCTION: "Nevertheless, in wide field of view instruments, as those imaging an extended source covering most of the space in front of the antenna, there are ..."

3.  “alias free” region:  It is the understanding of this reader that once aliasing is present and the scene fills the field of view of the antenna, there is no truly “alias free” region.  Even a very good antenna array observing at nadir will have error due to aliasing if the spacing is not less than 0.5 wavelengths.  How is SMOS’s “alias free” region different?

RESPONSE: THE WHOLE SPACE IN xi,eta IS THE UNIT CIRCLE. DUE TO VISIBILITY SAMPLING, THIS UNIT CIRCLE IS REPLICATED (ALIASED) AT DIFFERENT LOCATIONS IN SPACE. WHEN THE MINIMUM SPACING IS 0.5 WAVELENGTHS [lambda/sqrt(3) FOR HEXAGONAL SAMPLING], THESE REPLICAS ARE TANGENT TO THE ORIGINAL CIRCLE, SO ALL THE SPACE IS ALIAS-FREE. IF THE SPACING IS INCREASED THE ALIAS CIRCLES OVERLAP WITH THE ORIGINAL ONE, BUT THERE MAY BE STILL AN AREA WHERE THERE ARE NO ALIASES, AND THIS IS THE SITUATION OF SMOS. IF SPACING IS FURTHER INCREASED, THEN THE ALIAS FREE REGION IS REDUCED AND COLLAPSES WHEN THE SPACING IS ONE WAVELENGTH [2/sqrt(3) FOR HEXAGONAL]

4.  “extended alias-free region”:  Lines 289-291 appear to be the only definition of this term, but they are not clearly written.  The phrase “which conform the” probably involves a typo.

RESPONSE: IT IS A COMMON TERM IN SMOS AND IS DEFINED BY USING THE REPLICAS OF THE EARTH DISK INSTEAD OF THOSE OF THE UNIT CIRCLE. THE SENTENCE HAS BEEN CHANGED TO IMPROVE CLARITY. NOW IT IS: "... regions where the earth aliases do not enter into the hexagon, which is the so-called ``extended alias-free field of view'' ...". AGREE THAT "CONFORM" IS NOT THE RIGHT WORD.

5.  Averaging:

a)  Lines 103-112:  The subject of the text is averaging rows in the matrix expression  V = GT with redundant spacing, but the wording is awkward.

RESPONSE: YES, THIS IS THE SUBJECT. TEXT HAS BEEN CHANGED. HOPE IS CLEARER.

b)  It is not clear to this reader that Eqn 7 can be used (line 112) “without any loss of information”.  Averaging in the manner suggested makes sense but it is a choice of options and it remains to be demonstrated in the case of different antenna patterns and variation in receiver characteristics, that there is no loss of information and that it results in thermal noise reduction (Line 118-119).

RESPONSE: THIS IS TRIED TO JUSTIFY IN THE FOLLOWING PARAGRAPH [LINES 113 TO 119] WHICH IS SUMMARIZED AS FOLLOWS: ONLY ONE BASELINE OUT OF A SET OF REDUNDANT ONES IS NEEDED TO RECOVER A GIVEN SCENE. THEN, ASSUMING THAT ANTENNA PATTERNS ARE PERFECTLY KNOWN AND THAT RECEIVERS ARE PERFECTLY CALIBRATED, USING DIFFERENT REDUNDANT BASELINES WOULD PRODUCE THE SAME IMAGE, EXCEPT FOR THE THERMAL NOISE. IN OTHER WORDS, A REDUNDANT BASELINE IS NOT ADDING ANY NEW INFORMATION. IT IS JUST ANOTHER MEASUREMENT OF THE SAME VISIBILITY SAMPLE. IN ANY CASE, THE ITALICS OF THE SENTENCE "without any loss of information" HAS BEEN REMOVED.

6.  Lines 139-140:  “hermitic ones”  Should this be “conjugate ones”?

RESPONSE: CHANGED.

7.  Lines 295-296:  Is figure 4 the correct figure?  There are only two examples in Fig 4.

YES, IT IS THE CORRECT FIGURES. IT SHOWS TWO SMOS SNAPSHOTS, ONE FOR X-POL AND OTHER FOR Y-POL.

8.  Eqn 4:  It is not clear to this reviewer that adding phase shift information to the antenna patterns is not counting the displacement of the receivers in a correlation pair twice.  Please check that in deriving Eqn 1, the assumption is made that the antenna pattern is at the receiver (i.e. in coordinates local to the individual receivers).

RESPONSE: EQUATION (4) ILLUSTRATES THE CHANGE OF REFERENCE PHASE OF THE ANTENNA PATTERN FROM THE CENTER OF COORDINATES OF THE MEASUREMENT SETUP TO THE ANTENNA POSITION IN THE ARRAY. IN PRACTICE, THIS IS DONE BY THE ANTENNA MEASUREMENT TEAM, WHICH PRODUCES THE FINAL ANTENNA PATTERS F(XI,ETA) AT THE CORRECT POSITION, DIRECTLY TO BE USED IN EQUATION (2). AS YOU SAY, IN EQUATION (1) ANTENNA PATTERS ARE AT THE RECEIVER COORDINATES.